# *More PEAS Please!* Process Evaluation of a STEAM Program Designed to Promote Dietary Quality, Science Learning, and Language Skills in Preschool Children

**DOI:** 10.3390/nu17111922

**Published:** 2025-06-03

**Authors:** Virginia C. Stage, Jocelyn B. Dixon, Pauline Grist, Archana V. Hegde, Tammy D. Lee, Ryan Lundquist, L. Suzanne Goodell

**Affiliations:** 1Department of Agricultural & Human Sciences, College of Agriculture and Life Sciences, North Carolina State University, Raleigh, NC 27695, USA; jocelyn_dixon@ncsu.edu (J.B.D.); pgrist@ncsu.edu (P.G.); rklundqu@ncsu.edu (R.L.); 2Department of Human Development & Family Science, College of Health and Human Performance, East Carolina University, Greenville, NC 27834, USA; hegdea@ecu.edu; 3Department of Mathematics, Science, & Instructional Technology Education, College of Education, East Carolina University, Greenville, NC 27834, USA; leeta@ecu.edu; 4Department of Food, Bioprocessing and Nutrition Sciences, College of Agriculture and Life Sciences, North Carolina State University, Raleigh, NC 27695, USA; lsgoodel@ncsu.edu

**Keywords:** STEAM, food-based learning, Head Start, teacher professional development, process evaluation, intervention

## Abstract

**Background/Objectives**: Traditional nutrition education can increase children’s exposure to healthy foods, but preschool teachers face barriers such as limited time and competing priorities (e.g., kindergarten readiness). Integrating nutrition into other learning domains (e.g., science) has been identified as a potential solution. However, teachers need more professional development. We developed the *More PEAS Please!* program to support preschool teachers’ integration of food-based learning (FBL) and science, seeking to improve children’s science learning, language development, and dietary quality. **Methods**: In this pilot study, we used a mixed-methods process evaluation to assess the program in five Head Start centers (*n* = 23 classrooms) across three rural North Carolina counties. We collected teacher data via surveys and interviews. **Results**: A total of 24 teachers participated in the full intervention by attending a one-day workshop, completing at least one of four core learning modules, and implementing 16 food-based science learning activities in their classrooms. Teachers were Black/African American (81.1%) and 43.56 (11.89) years old. Teachers reported varying engagement levels and high satisfaction with the program, sharing increased confidence in FBL and science integration. However, barriers such as time, technology, and the coronavirus disease (COVID-19) limited full participation. **Conclusions**: Our findings suggest that the program is feasible and well received in Head Start settings and has promising impacts on classroom teaching practices. The findings will guide revisions to the PEAS program. Future research evaluating the revised program using a comparison group will be explored.

## 1. Introduction

Preschool-aged children (3–5 years) spend more than 30 h each week, and over half of their daily waking hours each day, in the care of early childhood educators [1,2,3]. Consequently, many early childhood teachers perceive themselves as “parents at school”, expressing a strong sense of responsibility and an intrinsic desire to support the health, happiness, academic readiness, and overall success of the children in their classrooms [4,5,6,7,8]. This perception is well founded as prior research supports teachers’ critical role in impacting both the health [9] and academic [10] outcomes of the children in their care.

Serving nearly one million children nationwide annually, Head Start is the largest federally funded preschool program in the United States that provides comprehensive early childhood education, health, nutrition, and family support services to children from families with limited resources [11]. Built on a “whole-child” approach, Head Start emphasizes both academic learning and health as foundational to a child’s development [11]. Therefore, Head Start teachers play a vital role in supporting the development of healthy eating habits and promoting school readiness in young children who are most at risk [9,10,12,13].

Traditionally, teachers have attempted to improve children’s dietary habits through classroom-based nutrition education, but how early childhood educators define nutrition education varies greatly [14]. Furthermore, teachers often implement nutrition education in a “stand-alone” or “siloed” fashion in preschool classrooms (e.g., “After we finish this literacy/math/science lesson, we will do our nutrition activity”), which may cause a decrease in how often nutrition education is provided overall [14,15,16,17]. For example, common nutrition education activities that teachers describe include teaching children about food colors, health benefits, food groups, portion sizes, food safety, and making healthy choices [14,15]. Even the language of the former Head Start policy, which required “nutrition education” to occur in the classroom at least once a month [18], conveys the notion that nutrition education is a “separate” task from regular classroom instruction for teachers to complete. Teachers also echo this separation, sharing that, while they value nutrition education, they face barriers like limited classroom time and competing priorities related to kindergarten readiness domains like math, science, or literacy [14,17,19]. Teachers and administrators have suggested that integrating nutrition education into other learning domains is a potential solution [14,17,19].

A promising integrative approach to nutrition education that may impact not only children’s diet but also their academic outcomes is food-based learning (FBL) [15,17,19,20,21]. Food-based learning has been defined as the “use of food as a tool to provide repeated exposure to improve children’s dietary behaviors and/or academic learning related to knowledge (e.g., science, mathematics, and literacy) and/or skills (e.g., gross motor, fine, physical)” [15]. By integrating traditional nutrition education into other learning domains (e.g., math, science, literacy), FBL removes the “silos” or “stand-alone” nature of traditional preschool nutrition education activities and instead integrates nutrition education into everyday classroom instruction. For example, children can learn about celery during a study on insects as an example of a plant that insects, such as butterflies, pollinate. This learning experience contrasts with simply making a butterfly out of celery sticks or learning that celery is green and contains Vitamin K [15].

A recent 2021 statewide study of 168 Head Start teachers in North Carolina (NC) found that 75% of teachers considered FBL very or extremely important in early childhood education [16]. Additionally, 66.1% believed that FBL was very or extremely important from the perspective of their center, state, and federal administrators [16]. Teachers reported being motivated to implement FBL in their classrooms to improve job performance, keep up with best practices, and better prepare children for kindergarten [16]. However, the teachers in this study echoed the same challenges faced by teachers in previous studies, including limited time [14,17,19], competing kindergarten readiness priorities, and the need for increased professional development opportunities [14,16]. For these reasons, there is a pressing need to explore the effectiveness of FBL interventions.

Science is an ideal learning domain to support content integration as it provides a developmentally appropriate and meaningful context for exploring key concepts, developing problem-solving abilities, and applying knowledge to real-world situations [22]. Children are naturally engaged during quality hands-on science exploration, which can support the development of new language, mathematics, health, and social skills [23,24]. For example, a teacher might read a book about apples growing to introduce a core life science concept [25]. Reading the book may teach children about healthy food and how plants grow and introduce new vocabulary. However, if the teacher uses the book with a hands-on science exploration, children could discuss what they already know about apples, conduct a taste testing experiment with different types of apples and/or predict and count the number of seeds inside, learn new words and concepts through reading about how apples grow, reflect on the importance of making healthy choices for our bodies, and ask more questions (e.g., how are pears similar to or different from apples?) to continue the cycle of inquiry [26].

Others have successfully integrated FBL and science education in this way. For example, the preschool–first-grade program We Inspire Healthy Eating (WISE) engages children in integrative learning throughout eight research-based discovery units. It uses a mascot, Windy the barn owl, who invites children to explore one target food a month through integrative activities and hands-on recipes [20,27]. One WISE integrative activity has children guess how many seeds they think will be inside a green bean pod. Children record their predictions and then open their pods and reflect on their predictions. Children then sort their pods from shortest to longest and describe the characteristics of the bean (e.g., green, smooth, bumpy, fuzzy, short, long). The activity ends with the children reading a book about how vegetable plants grow. Other programs, like Preventing Obesity by Design (POD), engage children in hands-on garden learning activities such as scavenger hunts to find seeds on the ground, followed by classifying them based on shape, color, size, or weight. Children then examine the parts of a plant and learn about their different functions. The activity ends with reading a book [28].

Studies among preschool teachers have indicated a need for more professional development opportunities to support teachers’ effective integration of FBL into other learning domains [14,15,29]. Similarly, research has also described preschool teachers’ need for more professional development opportunities in science education [30,31], pointing to a unique opportunity to address needs in both areas. Unfortunately, most studies published to date have been unable to evaluate professional development and instead have considered this component of their intervention as the background. Of the studies that have been conducted, few have provided useful findings. For example, one study explored the effects of a professional development workshop for preschool teachers focused on integrating science learning activities into their classrooms. However, after the workshop intervention, teachers reported that they did not implement any learning activities, citing a need for additional support following the initial workshop training. Unfortunately, the researchers did not collect process evaluation data that would aid in the further understanding of teachers’ support needs beyond the request for more professional development [32]. There is a critical need for more rigorously evaluated professional development programs that support teachers’ effective integration of nutrition and science, particularly professional development programs aligned with a specific curriculum [30].

In summary, preschool teachers are generally enthusiastic and capable of supporting early learning in nutrition and science, but need additional support to do so effectively [33,34]. In response to this well-described need, our team created the *More PEAS Please!* program. PEAS stands for Preschool Education in Applied Science; it is a multi-component intervention designed to improve teachers’ food-based science teaching practices and young children’s (3–5 years) school readiness by improving science knowledge, language skills, and dietary quality.

## 2. Materials and Methods

### 2.1. Study Aims

This paper reports on a process evaluation of the *More PEAS Please!* program, conducted during a pilot implementation involving Head Start teachers. Rather than emphasizing outcomes alone, process evaluation prioritizes an in-depth examination of implementation processes, focusing on the quality, consistency, and contextual factors influencing program delivery in early childhood educational settings. These aspects remain essential to fostering meaningful engagement and supporting long-term program success [35].

Using a mixed-methods approach that combined teacher surveys with in-depth interviews, we explored multiple facets of implementation. These included how closely the program followed its intended design; the degrees of teacher participation, engagement, and satisfaction; how teachers responded to the materials and training; contextual factors; and the kinds of challenges that emerged during the pilot phase [36].

This evaluation aimed to monitor program delivery and uptake of the program and generate insights for future refinement. The findings highlight both strengths in the implementation approach and areas where modifications may enhance engagement and long-term integration. This study contributes to the development of a more adaptable and scalable model for early childhood nutrition education, while also promoting transparency and accountability through detailed documentation of the implementation process.

### 2.2. More PEAS Please! Development and Program Overview

Following a needs assessment [15,16,37], our transdisciplinary team of faculty/staff and community partners with expertise in early childhood education, early science education, speech–language pathology, nutrition science, curriculum integration, and teacher development refined the *More PEAS Please!* program over 2 years. Uniquely, *More PEAS Please!* emphasizes improvements in teaching practices through teacher professional development versus reliance on a specific curriculum [21]. Evidence-based strategies ground each component of the intervention, including high-quality science learning through “Practice Science”, food-based experiences through “Engage the Senses”, language skills through “Apply Science Talk”, and overall learning support using developmentally appropriate strategies through “Support Learning” [21].

We designed the intervention to be implemented over a full school year, with initial teacher training occurring in pre-service training (mid-August before the school year), learning modules (beginning in November/December), and classroom implementation (January–April) (Table 1). The research team provided technical support throughout the school year. The program was to consist of four integrated components: (1) a 1-Day Kick-Starter Workshop; (2) six interactive, on-demand online learning modules; (3) classroom implementation; and (4) virtual Learning Communities (LCs). We provided teachers with all classroom materials, teaching guides, and instructions to implement the intervention effectively. The federally funded Child and Adult Care Food Program (CACFP) [38] provided partial funding to support perishable food items needed for classroom learning activities; our team purchased and delivered any remaining food and nonperishable supplies as required. Finally, we encouraged and supported the teachers in broadening the educational impact by involving families and the community in the learning process. We provided teachers completing the program with Continuing Education Units (CEUs) that could be applied toward annual training or licensure requirements.

#### 2.2.1. Component 1: One-Day Kick-Starter Workshop

The PEAS 1-Day Kick-Starter Workshop was held in August during pre-service training (before the school year). This workshop introduced teachers to the program’s mission and objectives, provided hands-on learning experiences centered on the PEAS teaching practices, and established center-based LCs. We challenged teachers to reflect on their current classroom practices during training, and opportunities were provided to learn how to apply the teaching strategies through a series of breakout sessions.

#### 2.2.2. Component 2: Interactive, On-Demand Online Learning Modules

Following the workshop, teachers engaged in ongoing professional development through six interactive online learning modules delivered through the PEAS Online Learning Platform housed on *Teaching Channel*. This multi-platform service delivers virtual teacher professional development [39]. While teachers could access online modules on demand, we also gave them a deadline to ensure that modules were completed before the related learning activities and LC meeting. Each module emphasized key elements of effective teacher training, such as active learning through practice, feedback, and reflection; coherence with Head Start program goals; a focus on child outcomes; and careful consideration of the duration, including contact hours spent in learning versus classroom practice activities [40,41]. We also designed learning modules to support teachers implementing the PEAS Practices and model science learning activities in their classrooms.

Module organization followed Head Start’s Early Learning Outcomes Framework for Effective Practice Guides, which focuses on teaching specific practices that support children’s development [42]: (1) Know—Modules provided teachers with whiteboard video-based content knowledge [43] focused on the teaching strategies that support each PEAS practice area; (2) See—The modules provided teachers with video clips of science learning happening in early education classrooms and asked them to reflect on the teacher–child interaction; (3) Do—The modules asked teachers to apply new classroom strategies through goal setting and implementing model science learning activities in their classroom; (4) Improve—At the end of each module, teachers were encouraged to reflect on their goals and discuss the new science learning strategies in the LC meetings. The PEAS team provided feedback on the goal setting and reflection exercises embedded within the modules.

#### 2.2.3. Component 3: Classroom Implementation

Online training can offer significant advantages in scale and cost [44,45]. However, training is most effective when it includes hands-on, experiential learning [46]. To support teachers’ professional development and ensure practical application, we provided each teacher with a PEAS Teaching Guide, which offered additional guidance on understanding and applying the PEAS Practices and included 16 model science learning activities to support teachers’ immediate classroom implementation. Teachers were asked to implement all activities in their respective classrooms over the school year using the calendars provided to them. We designed PEAS activities to be implemented during both large-group instruction, such as circle time, and small-group activities in classroom centers. Additionally, we aligned the PEAS Practices and learning activities with the Next Generation Science Standards (NGSS) [25] and Head Start’s Early Learning Outcomes Framework [42], ensuring compliance with established educational standards while addressing common barriers identified by teachers [13,14,29,47].

In alignment with the NGSS Life Science Disciplinary Core [25], we structured the 16 learning activities into four thematic units focused on life science topics: (1) living and non-living things; (2) seeds; (3) plants; and (4) plant parts. Each unit’s hands-on science learning activities had simple procedures and suggested conversation starters to facilitate teacher–child interactions. Each activity featured one or more PEAS target vegetables (carrot, tomato, spinach, and peas). In the final activity of each unit, children participated in a taste test featuring a recipe with one or more of the target vegetables. To further support repeated exposure to the target vegetables, we collaborated with Head Start Health Managers at each center to incorporate or increase the frequency of inclusion of the target vegetables in their six-week cycle menu. We selected these four target vegetables because they were locally available, affordable, and identified by Head Start families as familiar [17,48].

#### 2.2.4. Component 4: Virtual LCs

PEAS staff supported the formation and operation of virtual LCs, which focused on implementing new teaching strategies and offering a supportive environment to address teaching challenges. The collaborative nature of the LC makes it a powerful tool for teaching and learning [49,50]. There is no built-in hierarchy within the group; instead, the group’s motto is “learning for all”, where all teachers learn and assist each other to improve their teaching practices/efficacy, ultimately impacting student learning [49,50]. Our team provided a segment of dedicated training on a robust LC model during the Kick-Starter Workshop. Teachers also received a Supplemental LC Guide to aid in these efforts. Our LCs were initially intended to be teacher-led and held at the center level. However, the partnering Head Start program only allowed teachers to meet in LCs on unpaid teacher workdays or school holidays. Teachers expressed concern that they would struggle to attend center-based LC meetings in addition to other personal and professional obligations. For these reasons, one team member with experience in LCs led these meetings virtually on prescheduled days across all sites.

#### 2.2.5. *More PEAS Please!* Program Theory and Theoretical Framework for Teacher-Level Intervention

Our program theory posits that improving teachers’ teaching self-efficacy and pedagogical knowledge/skills will positively change classroom practices and improve child outcomes (dietary quality, science knowledge, language development). Bandura (1993) defined self-efficacy as “a person’s belief in an ability to succeed in a particular situation” [51]. Self-efficacy influences how individuals think, feel, behave, and motivate themselves. In teaching, self-efficacy refers to a teacher’s belief in their capacity to motivate children and promote learning effectively. Teachers’ self-efficacy is closely linked to their confidence in fostering positive children’s educational outcomes [52].

In support of our program theory, the theoretical framework is aligned with Social Cognitive Theory [53] and the Interconnected Model of Teacher Professional Growth [54]. The theoretical framework features four key domains that drive changes in teaching practices: external (introduction of new information and experiences), personal (teachers’ knowledge, skills, beliefs, and attitudes), practice (professional experimentation within and beyond the classroom), and consequences (impactful outcomes for teachers and students that the teacher values). The program’s professional development is designed to facilitate effective change by introducing new knowledge and skills through face-to-face workshop and online learning modules; enabling active learning and practice through the implementation of model classroom learning activities; and supporting opportunities for goal setting, peer collaboration, and reflective practice through LCs and technical support [40,41]. Key mediators of change include outcome expectations (the personal value that teachers assign to new practices), environmental influences (such as support from administrators and families), the active implementation of new practices, and reflection on individual beliefs and attitudes linked to these new practices (Figure 1).

We conducted the study in August 2021–June 2022, in collaboration with a partnering Head Start organization that operated five centers (23 classrooms) across three rural NC counties. We used an explanatory sequential mixed-methods research design aligned with key process evaluation methods to assess program implementation [55]. We collected data using surveys (baseline, formative, and post-intervention) and in-depth interviews. We leveraged quantitative data to examine key factors influencing participants’ perceptions of program implementation, feasibility, and acceptability, while qualitative data generated a more robust understanding of teachers’ experiences. East Carolina University’s Institutional Review Board (#21-001272) reviewed and approved all study materials.

All lead and assistant teachers employed by the partnering Head Start program were eligible to participate in the study (i.e., completing process evaluation surveys and an in-depth post-intervention interview). We recruited teachers to participate on the first day of the PEAS Kick-Starter Workshop. We gave teachers the option to “opt out” of participating in the study, and they were informed that deciding not to participate would not exclude them from receiving program-related training or resources. All teachers consented to participate. Teachers who participated in the study were eligible for compensation of up to USD 150 on a reloadable Greenphire ClinCard [56].

### 2.3. Data Collection and Measures

We employed process evaluation methods to examine the implementation of the *More PEAS Please!* program. Process evaluations enabled the team to determine whether interventions were implemented as planned, identify factors influencing execution, and provide context for the interpretation of outcomes [57,58]. Our process evaluation was guided by the Reach, Effectiveness, Adoption, Implementation, Maintenance (RE-AIM) framework [59] and the Saunders et al. framework [36]. Table 2 describes each process evaluation tool [36].

Before beginning the pilot study, we cognitively evaluated all survey-based process evaluation tools with one Head Start teacher per survey, recruited by center directors. Each virtual cognitive interview was approximately 30 min. For each question, we asked the cognitive interview participants about confusion, clarification, and general edits that they would give for each survey item [60,61]. We made minor changes to the survey tools, such as clarifying vague words like “people” to be more specific (i.e., “teachers”). In some cases, an additional option was added. For example, a question asking about factors that caused a teacher not to implement their PEAS learning activities originally had the response “*I did not allot time for them*”. The teacher reviewing shared that, while this may be the case, teachers often plan time for an activity but another obligation arises at the last minute. To address this, we added the option “*I ran out of time*”. We compensated teachers with a USD 10 gift card for their time.

#### 2.3.1. Kick-Starter Workshop Experience Survey (Baseline, Post-Workshop)

Teachers completed a researcher-developed 15-item, five-minute survey after the 1-Day Kick-Starter Workshop. The first nine questions assessed teachers’ experiences with the workshop (e.g., “*The workshop helped me understand the components of the PEAS Institute.*”). We collected responses using a five-point Likert scale, ranging from 1 = strongly agree to 5 = strongly disagree. The remaining six open-ended questions sought qualitative feedback on workshop effectiveness (e.g., “*What did you learn most from today’s training?*”).

#### 2.3.2. Online Learning Module Formative Surveys

Teachers completed formative surveys following the content of online learning modules 2–5. These assessments measured process evaluation outcomes such as fidelity, dose delivered, dose received (exposure and satisfaction), reach (participation), maintenance, and effectiveness. Following an explanatory sequential mixed-methods design, the responses from these surveys informed subsequent in-depth interviews addressing barriers, motivators, and facilitators related to program implementation [55].

#### 2.3.3. Post-PEAS Experience Survey (Post-Intervention)

Teachers completed a researcher-developed 21-item, 15 min survey at post-intervention. The first ten questions assessed teachers’ experiences with the PEAS Program (e.g., “*How likely are you to continue to use each of the PEAS Practices now that you have completed PEAS?*”). The next eight questions assessed teachers’ access to PEAS training materials (e.g., “*Which PEAS resource did you use most frequently to complete modules?*”). The last three questions asked teachers to rate the priority levels of science education and FBL in their classrooms, both from their perspective and as they perceived it to be prioritized by others—including coworkers, families, and Head Start administration at the local, state, and national levels.

#### 2.3.4. In-Depth Interviews (Post-Intervention)

We conducted semi-structured, in-depth telephone interviews with participating teachers to explore their experiences with the *More PEAS Please!* program, focusing on barriers, motivators, and facilitators of program implementation. We developed the interview guide to align with the RE-AIM framework [59]. An experienced research team collaboratively designed the questions, informed by the relevant literature and program components. To ensure personalized discussions, we reviewed the teacher responses from the formative surveys beforehand to identify implementation challenges and successes. A trained team member, independent of the implementation activities, conducted all interviews [62]. Prior to data collection, this team member pilot-tested the interview guide with two Head Start teachers [62]. We did not consider these individuals part of the official sample size. The team member recorded and transcribed the interviews using the Rev App recorder on university-owned electronic devices (e.g., iPad).

### 2.4. Data Analysis

We used descriptive statistics to analyze demographic information and summarize the data using the IBM^®^ SPSS^®^ software version 29.0 (IBM Corp., Armonk, NY, USA) [63]. We summarized demographic data and Kick-Starter Workshop data as frequencies (%), and items were merged across all formative surveys via means (standard deviations). For the qualitative analysis, a phenomenological approach guided both data collection and interpretation. Following established qualitative research practices [62], a trained team member reviewed the interview transcripts to identify key themes and construct a coding framework. We undertook a more detailed analysis, drawing upon the phenomenological approach and the RE-AIM framework, to examine recurring themes and sub-themes. This process involved describing participants’ experiences, identifying significant statements about their engagement with the program (horizontalization), categorizing these statements into thematic patterns, and ultimately developing a comprehensive interpretation through both textural (what was experienced) and structural (how it was experienced) descriptions [55]. We used several validation strategies to ensure the credibility and trustworthiness of our findings. These included peer debriefing and data triangulation with teachers by sending individual summaries to participating teachers for the verification of the interpretations. We also shared summarized results with program partners [64].

Table 3 reports the teacher demographics. A total of 42 teacher participants (23 lead teachers; 19 assistant teachers) attended the Kick-Starter Workshop, and we formally enrolled them in the study. Following the workshop training, approximately 57.1% (*n* = 24) of the teachers continued with program implementation, completing at least one core module by the end of the study. At post-intervention, 19 teachers had completed all intervention components (45.2% retention rate). Among teachers who dropped out of the study, 23.8% formally withdrew (i.e., 5 teachers left Head Start, 3 teachers moved to an Early Head classroom, and 2 teachers withdrew for personal reasons). The remaining 8 teachers (19.0% of the sample) did not complete any of the learning modules or formative assessments. Notably, one center, which faced significant challenges due to the negative impact of coronavirus disease (COVID-19) on staffing and program administration, employed all but one of these teachers (*n* = 7).

## 3. Results

### 3.1. Process Evaluation Outcomes by Program Component

This section describes the fidelity, dose delivered, dose received, and reach process evaluation outcomes by program component. Table 1 details how the program was implemented, and Table 2 details how each program evaluation measure aligns with the evaluation tools.

#### 3.1.1. Kick-Starter Workshop

Of the 42 teachers enrolled in the study at the Kick-Starter Workshop, 34 completed a Post-Workshop Experience Survey. The teachers rated the facilitators of the Kick-Starter Workshop highly for fostering engagement, with 100% of participants stating that they agreed or strongly agreed that the workshop encouraged interaction and effectively conveyed information. Additionally, 97% felt that the facilitators were knowledgeable, and 100% agreed or strongly agreed that the workshop provided a clear understanding of the four program components, outlined expectations for engagement throughout the academic year, and explained how to access professional development materials.

The open-ended post-workshop survey questions asked participants, “*Was the workshop what you expected?*” Most participants expressed a positive experience. One participant responded, “NO! It was way better. I had fun.” Another participant stated, “The workshop was a lot better than I expected. I loved how the information was presented and was interactive and hands-on.” Conversely, one participant expressed concern regarding the amount of information being presented in a single day, stating, “[I’ve] been in training all last week so it was a lot to take in for one day.” Teachers (*n* = 17) who participated in the post-intervention in-depth interviews reiterated these findings: “When they came out and did the little training on it, that was great. I enjoyed the different activities” (Teacher 4).

#### 3.1.2. Learning Modules

Following the workshop, teachers gained access to six learning modules (Table 1). Formative assessments only accompanied core modules (modules 2–5). Of those who attended the workshop, 19 teachers (46.3%) completed all four core teacher learning modules, and an additional five teachers completed at least one of the teacher learning modules (58.5%). Teachers completed formative assessments at the end of each of the four core learning modules (Table 2), which asked about their perceptions of the module’s effectiveness (e.g., “*This module presented new information to me that I did not know before*”; five-point Likert scale, 1 = strongly agree to 5 = strongly disagree). To facilitate a clear interpretation of the participants’ responses and to emphasize their levels of agreement with key statements, when calculating the learning module scores, we re-coded the item responses into dichotomous variables, and we assigned “strongly agree” and “agree” a value of 1 and “neutral”, “disagree”, and “strongly disagree” a value of 0. We computed mean scores for each item, with higher mean values reflecting a more positive response. Overall, teachers reported the core learning modules to be effective in delivering novel content (M = 0.59, SD = 0.34), and teachers viewed the modules as beneficial in fostering science teaching practices. Finally, participants reported that the module contributed to developing their scientific practices (M = 0.89, SD = 0.26).

The assessments also collected data on barriers and supports to implementation using “*select all that apply*” items. Barriers included time, resources, and administrative or technical difficulties. Supports included staff, PEAS team, administrative, or technology support and/or sharing ideas with other teachers. We coded the responses dichotomously for barrier and support items, with 1 indicating that participants selected an option and 0 indicating that they did not. Mean scores represent the average proportion of respondents who endorsed each response. The most frequently reported support that participants experienced was “*Knowledge gained from learning modules*”, with a mean of 0.64 (SD = 0.35). Additional sources of support included staff assistance during instructional activities (M = 0.59, SD = 0.40) and assistance from the PEAS team in acquiring the necessary materials (M = 0.45, SD = 0.40).

Participants generally regarded the modules as of high quality and easy to comprehend. We assigned the item “*This module was easy to follow*” a mean score of 0.86 (SD = 0.26), which suggested that most respondents agreed or strongly agreed that the instructional materials were clear. The interview participants concurred: “[The learning modules were] easy to work through and I enjoyed learning different stuff about science and expanding knowledge on it” (Teacher 1).

Although all participants (100%) initially felt confident using the *Teaching Channel* platform, technical assistance data later revealed that many struggled with navigating technology and internet access challenges. In response, we created paper-based modules with content that was identical to the online version, with training videos accessed via QR codes. Teachers could choose between the online platform and the paper-based option. By the end of the program, teachers had completed 72% of the learning modules using the paper module version and 28% using the online platform *Teaching Channel*.

Following the intervention, participants described technological challenges with the online training modules. Some were uncomfortable navigating the platform, while others faced unreliable internet connectivity at school. As a result, some teachers felt compelled to complete the training on their personal devices at home: “I couldn’t do it at school, because [I] couldn’t get on the internet. So basically I had to use my telephone. And of course, they don’t like phones out on the job. So we did what we had to do” (Teacher 9). Accessing Head Start email accounts also posed challenges: “For one thing I couldn’t get in my email. And I still can’t get into my work email. I gave her [PEAS staff] my personal email, because I don’t know what’s going on with my email” (Teacher 10). Because email was an unreliable form of communication, these barriers negatively affected participation as we primarily sent reminders and updates through email.

#### 3.1.3. Learning Activities

During the Kick-Starter Workshop, we provided teachers with a PEAS Teaching Guide, including four model science learning activity units to support the implementation of new teaching strategies in their classrooms. Following the Preschool Cycle of Discovery, each unit featured four activities aligned with an inquiry-focused question (i.e., What do we know?; What do we wonder?; What more do we want to know?; and What have we learned?) for a total of 16 activities. As part of the core modules (modules 2–5), teachers applied learned knowledge into the classroom through the four PEAS learning units (16 activities) implemented over approximately 16 weeks (Table 1). A formative assessment was administered after each core module and asked teachers, “*Which of the following model learning activities did you teach in your classroom this month?*” Teachers most frequently reported delivering Activity 1, “What do we know?”, with slightly lower but comparable implementation rates for Activity 2, “What do we wonder?”; Activity 3, “What more do we want to learn?”; and Activity 4, “What have we learned?” (Table 4). Notably, only one participant reported not teaching any learning activities in their classroom during one of the core modules. These findings suggest that the core components of the instructional framework were widely delivered in classrooms during the reporting period, indicating the strong uptake and delivery of the intended curriculum.

The teachers in the qualitative interviews echoed these sentiments: “[before PEAS] I talked about plants, [but] we don’t have food to go along with it. So having the different foods to go along with talking about the tomato, how different parts of the tomato, or how the tomato grows… They could feel it, taste it... They used all the senses” (Teacher 4).

Other teachers appreciated that the PEAS activities supported reading books about food and encouraged role modeling of healthy eating inside and outside mealtimes, encouraging children to try new foods. One teacher shared,

If they’re not used to it, they won’t eat it. So, we try to encourage children to eat. Sometimes reading books helps children to open up more to say, ‘Oh, I’m going to try this. I’m going to try this food.’ And so, by reading those books and then getting the teachers to model that, ‘I’m going to try that too. I don’t like that either, but I’m going to try that food as well.’ So, I think that helps a lot with the children, getting them involved in their eating.(Teacher 5)

Other teachers shared that they enjoyed the activities but the children seemed tired of the same four target vegetables, suggesting that future versions of the program increase the variety and potentially include fruits: “The PEAS project was good, I would just say, sometimes maybe we could get more interesting foods. Maybe if we could mix the vegetables and fruits together” (Teacher 16). Based on the Head Start enrollment data, the intervention exposed approximately 460 children within the 23 participating classrooms to a maximum of 16 model science learning activities. However, collecting child attendance data for activities was not feasible at the time of this study due to complications associated with COVID-19 (e.g., children frequently transitioned between classrooms due to staff shortages or quarantine outages).

#### 3.1.4. LCs and Other Supports

Our team established virtual LCs across all centers to support ongoing professional growth and PEAS progress, inviting educators to meet two times throughout the school year. At the first LC meeting (February 2022), five teachers attended (three from County 1, two from County 2); three teachers (three from County 1) attended the second meeting (April 2022). Teachers who were unable to participate in the LC meetings stated their barriers in the qualitative interviews: “the day they had the meeting, the Zoom meeting, we was off of work that day [in-service]... I was at a doctor appointment… So I really didn’t even know nothing about that they was having a meeting until a coworker had called me and that’s why I wasn’t on it” (Teacher 11). Another teacher shared, “I don’t have a working laptop at home and my phone at the time would only hold so much information and it would stop and I couldn’t get on. And the internet that is around in my house, because I live in a little rural place, sometimes you can’t get on” (Teacher 17).

Teachers who were present during the LC meetings shared positive PEAS experiences, stating that they had started to teach science more and enjoyed “seeing the excitement on children’s faces”. Teachers also shared that positive role modeling seemed to reduce children’s resistance to trying new foods. Other teachers reiterated that, while time was initially a barrier to PEAS, it became less of a barrier as they became more familiar with the program.

Finally, to provide teachers with support outside of scheduled LC meetings, we also facilitated monthly technical assistance check-ins to reinforce their learning and address challenges. By the end of the pilot, all participating educators had access to the full professional development, resources, and ongoing guidance, providing opportunities to apply PEAS in their teaching practices and classroom learning.

### 3.2. Overall Program Satisfaction

Teachers reported high levels of satisfaction with the program, noting its role in enhancing their instructional skills, increasing the opportunities to engage in science in the classroom, and demonstrating how PEAS strategies could be integrated across various learning domains. Many teachers also expressed general satisfaction with their program experience.

Among the interviewed teachers (*n* = 17), more experienced teachers shared that the program reinforced the best teaching practices that they had previously used but had stopped using. One teacher reflected, “Even though I taught a long time, I had forgotten…Because most questions that I have a tendency to ask the kids …they can give me a yes or no. But when you ask open ended questions, it gives their mind time to really think” (Teacher 14). This same teacher shared that she “enjoyed participating in PEAS” and “even tried to do more” science in the classroom “because it was hands-on and the information that was in the kit was right on target” (Teacher 14).

Another teacher shared that the program “gave me the opportunity to apply more science in my classroom, where in the past I was not able to.” She explained that PEAS helped her to overcome previous barriers, such as “not having the materials and not really knowing where to start, like with what activities to start with, what to say, how to start, how to explain. The PEAS program provided ways for us to break it down so that we, as well as the children, can understand it” (Teacher 13).

When reflecting on the program, the interview participants shared that the PEAS materials sparked deeper thinking and new excitement about science. One participant shared, “Honestly, when I was in school, science was my weakest subject. I couldn’t stand science. And I just found the overall [PEAS] session was just exciting. It got my brain thinking and going more” (Teacher 2).

Participants also provided specific examples of how they were excited to apply the PEAS Practices in their classroom. One participant explained how she used WH words, a practice found in the *Apply Science Talk* module, to ask the children in her class open-ended questions:

Let’s talk about this plant. And you just go from there. “Tell me what you see?” “What do you think it smells like?” It might be collard greens. “Collard greens are all green” [children say]. “How do you know?” If you ask them one question, it’s almost like scaffolding. You continue to ask open ended questions to where there’s no “no or yes” to it.(Teacher 12)

This same teacher explained that, over time, she saw children’s science knowledge change through their participation in PEAS: “[At first] the kids were like, ‘I don’t know what that is.’ Then we go back and ask them, maybe two weeks later…then they would remember, ‘oh, that’s the stem, that’s the root, oh, we need sunlight and water’” (Teacher 12).

Lastly, teachers expressed general satisfaction with the PEAS program: “I liked how PEAS broke everything down… The modules were great. The activities, the way it was laid out and planned out for us, it really was simple. I don’t think it could have been any better because I think it was really great” (Teacher 13). Another teacher shared that the program positively impacted her as a teacher, stating,

If I had to create a new [science lesson], I guess I feel like I will be better prepared to apply the strategies that I’ve learned…without overthinking the process of teaching science. I’ve been working with children for almost 10 years and science just never was my go to. But now that I’ve actually had experience with PEAS, I feel like I would be more willing and capable of actually teaching science to young children.(Teacher 2)

Despite positive reports, some teachers reported decreased satisfaction due to obstacles to participation. Across the formative assessments, teachers reported the most prevalent challenge as a “*lack of time in our schedule to increase the amount of education on this topic*” (M = 0.40, SD = 0.37). All interviewed teachers mentioned a lack of time as a barrier to completing training and/or implementing PEAS in the classroom. One teacher shared, “PEAS was basically done on my time at home” (Teacher 14). Finally, teachers also saw limited knowledge about integrating the life science topic into other regular activities (M = 0.20, SD = 0.07) as a challenge. Nevertheless, teachers felt that PEAS was important:

As a teacher and dealing with young children, I think the PEAS project is a much needed program, because we look at some of the children that we serve in our programs. They may not be able to be introduced to the different vegetables… [or] just planting a flower, being in their home environment. Because as a teacher, you never know what a child faces at home. When they come to school, we are like a safe haven. We love on them, and we help nurture and teach them different things. So I think this project brings everything together.(Teacher 16)

### 3.3. Maintenance

Many teachers reported that they were likely to continue to use the PEAS Teaching Strategies, stating that they were highly likely (HL) or likely (L) to continue using the teaching strategies *Practice Science* (HL, 55.6%; L, 44.4%), *Engage the Senses* (HL, 61.1%; L, 38.9%), *Apply Science Talk* (HL, 61.1%; L, 38.9%), and *Support Learning* (HL, 55.6%; L, 44.4%) (highly unlikely (1) to highly likely (5)). Additionally, most teachers reported being likely or highly likely (83.3%) to recommend PEAS to a coworker. Teachers named the PEAS teaching videos (27.8%) and the PEAS Teaching Guide with model science learning activities (27.8%) as the top resources that they would continue using. Teachers rated the online learning modules (33.3%) as the least helpful. Teachers rated access to PEAS resources most valuable (38.9%), followed by new science materials (33.3%), gift cards (16.7%), and food for learning activities (11.1%).

We asked the teachers to imagine their classrooms in five years and share which PEAS Practices they would still be using and which they would leave behind. Teachers shared, “All of them because they all go together if you do it right. I can’t see you having one without the other because you’re going to miss out on something if you let one go” (Teacher 7). Another teacher said, “Science is in your everyday life. And it’s something that children and teachers need to know. So I think I would love to use [PEAS] in the near future because you can never stop learning about science” (Teacher 3).

### 3.4. Contextual Factors

Finally, we documented contextual factors, including COVID-19 and administrative support, as key external influences on program outcomes and future maintenance. During COVID-19, participating Head Start centers had a policy requiring a one-week classroom closure if a child tested positive, requiring centers to send all teachers and children home. If staff reported another case during this quarantine, the closure was extended by an additional week, totaling two weeks. We were only able to collect quarantine data from teachers who completed formative surveys, which revealed that, during Module 2 (January), 33.3% of teachers reported at least one week of quarantine, with two classrooms closed for two weeks. For Module 3 (February), 13.6% of teachers reported one week of quarantine, with one classroom closed for two weeks. Teachers reported no quarantines during Module 4 (March) or Module 5 (April). Secondly, the interviewed teachers emphasized that support, or a lack thereof, from center administration was key to their PEAS success. For example, teachers shared positively, “Our center supervisor, she’s one that encouraged us to participate [asking us] ‘Have you done your PEAS today?’” (Teacher 5). On the contrary, other teachers shared more critically, “[My] supervisor came in to me, she was kind of rude. And she was like we don’t have time to do that” (Teacher 14). Other teachers expressed neutral feelings about the role that their center supervisor played in their program participation: “I wouldn’t say like the director, everybody from the center motivated me to participate. It was mostly the PEAS people” (Teacher 8).

## 4. Discussion

In this pilot study, we used a mixed-methods approach to examine the implementation, feasibility, and acceptability of the *More PEAS Please!* program [55]. The findings suggest that the PEAS Program is feasible and acceptable within a Head Start setting as 100% of teachers reported that the Kick-Starter Workshop adequately prepared them for the PEAS program, and most found the modules easy to follow. Of the teachers who completed the full program, 89.0% of teachers reported improvements in their science and nutrition teaching practices. During the interviews, teachers expressed surprise at their children’s ability to comprehend and “do” science, which may reflect their previous beliefs and low self-efficacy about science education in early childhood [65].

During the qualitative interviews, teachers expressed that PEAS significantly boosted their confidence in teaching science. Before PEAS, many reported little to no self-efficacy in this area [16,22,66]. These findings align with Bandura’s theory of self-efficacy development, wherein positive experiences throughout PEAS contributed to increased teacher confidence [67]. In alignment with our theoretical framework, prior research further supports that professional development in science instruction may bolster teaching confidence and improve science education teaching practices [68,69,70,71]. This is critical as quality science experiences help children to build a sense of identity that they can “do” science, which may influence their future interest in a career in science, technology, engineering, arts, and mathematics (STEAM) fields [22,30,72].

While much of the feedback was positive, several challenges emerged as this pilot took place in the aftermath of the COVID-19 pandemic (2021–2022). Prior research indicates that, even before the pandemic, Head Start teachers were already at risk of burnout due to suboptimal working conditions, low wages, job dissatisfaction, and high turnover rates—factors also linked to increased depression, inactivity, and poor diet quality [73,74,75,76]. These conditions are reflected in the high employee turnover rates seen in early childhood education before COVID-19, with 30–46% of all early childhood teachers leaving the profession within their first 3 years of teaching [77]. In March 2020, the onset of the COVID-19 pandemic exacerbated these challenges, as teachers faced additional stressors related to navigating the complexities of pandemic-era early childhood education [78,79]. For example, one study conducted in the northwestern United States suggested that 57% of early childhood educators reported high levels of stress during COVID-19 due to workplace closures/changes and economic insecurity [80].

COVID-19 also had tremendous negative impacts on teachers’ overall daily lives, including increased food insecurity, unemployment, anxiety, depression, and stress [81,82,83,84], with these effects even more prevalent in adults from low-resource backgrounds, such as many Head Start teachers [75]. After the return to “normal” teaching (e.g., in-person learning), studies suggest that preschool and primary-grade teachers experienced almost three times the prevalence of anxiety compared to the general population, due to the perceived high risk of contracting COVID-19 from young children who were still mastering good personal hygiene practices (e.g., covering cough, washing hands) [85,86,87]. Studies also suggest that teacher engagement decreased during COVID-19 and the years following, making teachers less likely to want to participate in “one more thing”, such as professional development [82,88,89], or respond to survey/research follow-ups [90].

We observed this pandemic-exacerbated stress and lack of engagement in our pilot study. We trained 42 teachers at the PEAS Kick-Starter Workshop (August 2021), from five centers across three Eastern NC counties. However, by the end of the intervention (May 2022), only 45.2% (*n* = 19) of the initially trained teachers had completed the full program, and an additional 14.3% (*n* = 6) had completed at least one module. Teachers who were partial or full implementers may have been from centers with more supportive administrations that encouraged their participation in the program [14,16]. For example, in the needs assessment for the PEAS program, teachers who perceived that FBL was a higher priority for their center administrators reported FBL as a higher perceived priority for themselves, drawing the conclusion that Head Start administration should demonstrate their prioritization of FBL as a model for teachers [16].

The remaining 42.9% (*n* = 18) of teachers only participated in the Kick-Starter Workshop and did not complete any core learning modules. Notably, 16 of these teachers were from the same center, which faced significant challenges exacerbated by COVID-19. These included staff shortages (often resulting in staff shuffling children between classrooms on a day-to-day basis), frequent and unexpected quarantines, mid-year staff hires, and administration turnover [91]. Additionally, we hypothesize that teachers who did not complete the program may have lacked the administrative support that other teachers at different centers received. Prior research supports the notion that early childhood teacher disengagement and turnover rates are higher when teachers perceive administrative support as poor [92,93,94]. Other challenges included limited time to complete program tasks and uncertainty about integrating PEAS activities into the existing classroom curriculum (i.e., Teaching Strategies Gold^®^). Future implementations should consider providing teachers with more direct training and resources to outline how PEAS activities can be integrated into their curriculum.

The formative assessment results suggest that “knowledge gained from PEAS modules” and “assistance from PEAS staff” were among the strongest supports for teachers. The learning modules were initially designed for online delivery, based on prior research suggesting online professional development as a viable solution for quality professional development and reducing barriers such as time, inconvenient locations, affordability, and transportation [95,96,97,98,99]. In addition, during our needs assessment (2020–2021), partner Head Start teachers expressed a desire for online professional development, which further informed our decision to base the modules online [16]. However, in reality, teachers faced barriers such as limited internet access, low technology proficiency, and an inability to use their phones at work [100,101]. To address this, our team converted all learning modules to a paper-based format shortly after the pilot began, with 62.5% of teachers using paper-based modules by the end of the program. Future implementations should consider providing more training on navigating the online platform and continue to provide paper-based options for teachers. Other nutrition education programs similarly concur that teachers need extensive time to familiarize themselves with a program’s learning platform and resources, even outside of a pandemic context [102]. Future iterations of PEAS should consider narrowing the number of practice areas introduced per year to allow for deeper focus and more manageable implementation.

Lastly, our team supported a virtual LC that extended across all participating Head Start centers. The goal was for teachers to meet regularly (e.g., during naps, before/after school, on in-service days) to share ideas, brainstorm, and problem-solve together as they progressed through PEAS [49,103,104]. However, teachers struggled to meet and desired more formal guidance. While not the original vision for the LC, our team adapted before the intervention began by helping to facilitate LC meetings to ensure that teachers had a dedicated space for collaboration. This shift aligns with prior research suggesting that preschool LCs often require additional support to be effective and sustainable [105,106]. More research is needed to determine which supports are necessary to successfully implement teacher-led LCs in early childhood settings.

While we originally intended to offer technical assistance on an as-needed basis, early feedback led our team to introduce a more formal, one-on-one approach. Starting shortly after the Kick-Starter Workshop, one of our team members visited each center monthly during naptime to provide individualized assistance, answer questions, and offer support. Other professional development programs in early childhood education have also seen success using a similar approach [107]. In the qualitative interviews, teachers shared that technical assistance was critical to their success. Future dissemination should explore sustainable strategies for providing support, such as the creation of a more formalized coaching model or partnership with Cooperative Extension Agents, to promote teacher retention, satisfaction, and success, potentially considering the work in [16,102].

### Strengths and Limitations

This study has several strengths. The *More PEAS Please!* program represents an innovative approach to integrating FBL with early science and language education, addressing both dietary quality and academic preparedness within the Head Start setting. We designed the intervention using a theory-driven framework, incorporating evidence-based teaching practices to enhance early childhood science instruction while increasing exposure to healthy foods. Additionally, this study employed a rigorous mixed-methods approach, leveraging both qualitative and quantitative data to assess feasibility and acceptability.

Despite these strengths, we acknowledge several limitations. As a pilot study, the sample size was small and limited to a specific geographic area, restricting its generalizability to broader populations. The study design also relied on teachers to complete formative assessments to ascertain certain data (such as quarantines), which limited data collection, particularly among the 18 teachers who attended only the Kick-Starter Workshop but did not complete additional program components. These pandemic-related constraints further complicated program implementation and data collection, highlighting the unpredictable nature of external factors in early childhood settings. Future research should explore strategies to improve program scalability and sustainability, particularly in low-resource early education settings. Additionally, implementing a comparison group in subsequent evaluations would strengthen the findings regarding program effectiveness. Despite these limitations, the findings from this pilot study provide valuable insights for the revision and expansion of the PEAS program, offering a promising model for the integration of science learning with food-based education in early childhood settings.

## 5. Conclusions

The *More PEAS Please!* intervention aims to improve children’s academic and dietary outcomes by supporting changes in teachers’ instructional practices related to early science learning. Providing teachers with professional development in both science and FBL has been suggested to promote a more supportive environment for children’s learning [68,69,70,71,108]. This study quantitatively and qualitatively explored teachers’ experiences with the PEAS pilot through process evaluation measures. The findings highlight potential areas for revision in future iterations of the PEAS program, such as changes to the program design and included components, program delivery, and assessment measures. This study also provides applicable lessons for the implementation of Head Start-based interventions in the aftermath of a pandemic. While future studies evaluating the potential impact of PEAS on teacher outcomes should be assessed using a comparison group, the teacher outcomes from this study are promising and may encourage educators to implement similar integrative STEAM approaches to enhance dietary quality while also positively impacting science learning and language development, potentially redefining what “nutrition education” looks like in the early childhood space.

## Figures and Tables

**Figure 1 nutrients-17-01922-f001:**
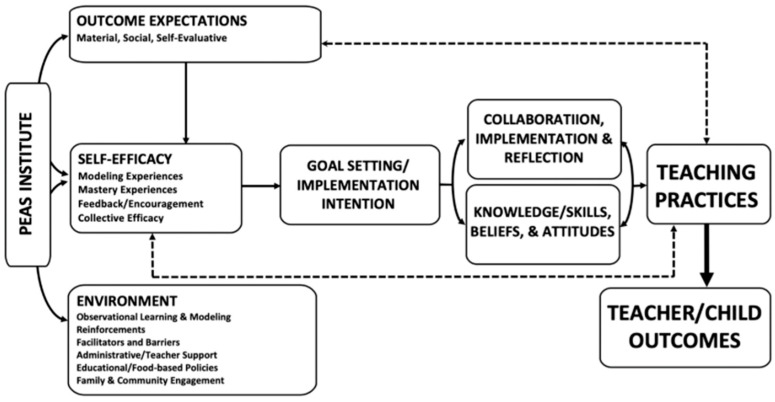
*More PEAS Please!* Teacher Theoretical Framework.

**Table 1 nutrients-17-01922-t001:** Timeline of *More PEAS Please!* program implementation.

Timeline	Activity ^a^	Activity Description
August	Kick-Starter Workshop	The PEAS 1-Day Kick-Starter Workshop introduced teachers to the program’s mission and objectives, provided hands-on learning experiences centered on PEAS teaching practices, and introduced the concept of LCs.
	Data Collection	Post-Workshop Experience Survey
November– December	Orientation Module Online Learning Module 1: Introduction to *More PEAS Please!*	Teachers completed the Online Orientation on how to how to use the *Teaching Channel* learning platform. Next, teachers completed Module 1, which provided an overview of the program and reviewed key concepts covered in the 1-Day Kick-Starter Workshop, including the importance of early science learning, the PEAS Practices, and tips for successful LCs. Teachers were prompted to “make a plan” for spring implementation.
	Technical Assistance Visit	Teachers met in person with a PEAS staff member to check-in on their learning and address any challenges.
January	Online Learning Module 2 ^b^*:* “S” is for Support Learning	Module 2 ^b^ introduced the PEAS Practice area Support Learning and related teaching strategies: (1) Using Effective Verbal Praise, (2) Enthusiastically Role Modeling, and (3) Encouraging Peer Collaboration.
	Technical Assistance Visit	Teachers met in person with a PEAS staff member to check in on their learning and address any challenges.
	Classroom Implementation	Teachers implemented “S” teaching strategies and the Teaching Guide’s Unit 1: Living and Non-Living Things model science learning activities in their classrooms.
	Data Collection	Online Module Survey ^d^
February	Online Learning Module 3 ^b^: “P” is for Practice Science	Module 3 ^b^ introduced the PEAS Practice area Practice Science and related teaching strategies: (1) Engaging in the Process of Science, (2) Learning the Big Ideas, and (3) Using the Tools of Science.
	Technical Assistance Visit	Teachers met in person with a PEAS staff member to check in on their learning and address any challenges.
	Classroom Implementation	Teachers implemented “P” teaching strategies and the Teaching Guide’s Unit 2: Seeds model science learning activities in their classrooms.
	Virtual LC Meeting #1 ^c^	Teachers met virtually in the LC to discuss the PEAS learning strategies, reflect on last month’s classroom activities, and set goals for upcoming classroom activities.
	Data Collection	Online Module Survey ^d^
March	Online Learning Module 4 ^b^: “E” is for Engage the Senses	Module 4 ^b^ introduced the PEAS Practice area Engage the Senses and related teaching strategies: (1) Exploring with the Senses, (2) Experiencing Culturally Relevant Vegetables, and (3) Benefiting from Repeated Exposures.
	Technical Assistance Visit	Teachers met in person with a PEAS staff member to check in on their learning and address any challenges.
	Classroom Implementation	Teachers implemented “E” teaching strategies and the Teaching Guide’s Unit 3: Plants model science learning activities in their classrooms.
	Data Collection	Online Module Survey ^d^
April	Online Learning Module 5 ^b^: “A” is for Apply Science Talk	Module 5 ^b^ introduced the PEAS Practice area Apply Science Talk and related teaching strategies: (1) Using Child-Friendly Definitions and Modeling Descriptive Words, (2) Asking Fair WH- and Open-Ended Questions, and (3) Revoicing and Restating Children’s Ideas.
	Technical Assistance Visit	Teachers met in person with a PEAS staff member to check in on their learning and address any challenges.
	Classroom Implementation	Teachers implemented “A” teaching strategies and the Teaching Guide’s Unit 4: Plant Parts model science learning activities in their classrooms.
	Virtual LC Meeting #2 ^c^	Teachers met virtually in the LC to discuss PEAS learning strategies, reflect on last month’s classroom activities, and set goals for upcoming classroom activities.
	Data Collection	Online Module Survey ^d^
May	Online Learning Module 6: Celebrate Your Success & Sustaining PEAS in Your Center	Module 6 celebrated teachers’ success and provided teachers with direction and support for sustaining the use of PEAS Practices in their classrooms, including continuing LCs independently. We prompted teachers to “make a plan” for the next school year.
	Data Collection	Online Module Survey ^d^; Post-PEAS Experience Survey; In-Depth Interviews

PEAS: Preschool Education in Applied Sciences; LC: Learning Communities. ^a^ Activities are presented in the order in which teachers completed them each month. The core intervention began with Module 2, which marked the start of classroom learning activity implementation. This phase lasted approximately 16 weeks and included Modules 2 through 5. Modules 1 and 6 did not include learning activities or formative assessments, as these assessments focused specifically on the implementation of learning activities. ^b^ Indicates core modules consisting of goal setting, classroom activity implementation, and reflection activities. Teachers only completed formative assessments for these modules. ^c^ In LC meetings, we encouraged teachers to discuss lessons learned from assigned modules (e.g., What strategies from the module have you learned and used in your classroom?), reflect on learning activities after implementation, and discuss children’s engagement and learning progress. ^d^ Online Module Surveys were considered formative assessments and were embedded in each online learning module. Teachers were asked to submit these electronic surveys after they had completed the assigned online learning module, classroom activities, and virtual LC meetings.

**Table 2 nutrients-17-01922-t002:** *More PEAS Please!* process evaluation measures ^a^.

Process Evaluation Element	Measure	Time of Collection
**Fidelity ^b^ (quality):** Extent to which the intervention was implemented as planned.	Post-Workshop Survey	Baseline, Post-Workshop
	In-Depth Interviews	Post-Intervention
**Dose Delivered (completeness):** Extent to which teachers received all program components, including the training provided through attendance at the Kick-Start Workshop and completion of the online learning modules.	Online Module Surveys	Formative, Ongoing
	In-Depth Interviews	Post-Intervention
	Administrative Data **^c^**	Formative, Ongoing
**Dose Received (exposure):** Teachers’ reactions to the program components and the extent to which teachers participated in follow-up components, including implementing classroom activities and engaging in center-based Learning Communities.	Online Module Surveys	Formative, Ongoing
**Dose Received (satisfaction):** Teachers’ reported satisfaction with the program, interactions with staff, and study team.	Online Module Surveys	Formative, Ongoing
	In-Depth Interviews	Post-Intervention
**Reach (participation rate):** Proportion of teachers that attended/completed each program component and reasons for choosing not to participate in them and not to participate in the intervention (e.g., supports, barriers)	Online Module Surveys	Formative, Ongoing
	In-Depth Interviews	Post-Intervention
**Maintenance ^d^:** Extent to which a program or intervention is sustained over time at both the individual and organizational levels	Post-PEAS Survey	Post-Intervention
	In-Depth Interviews	Post-Intervention
**Contextual Factors:** Environmental support or challenges unrelated to the intervention that may impact intervention implementation and outcomes	Online Module Surveys	Formative, Ongoing
	In-Depth Interviews	Post-Intervention

^a^ Adapted from Saunders et al., 2005 [36]. ^b^ Due to coronavirus disease (COVID-19), we had limited access to the classroom. Therefore, we assessed fidelity using a combination of surveys and in-depth interviews versus traditional direct observation. ^c^ Administrative data were collected by the research team throughout the school year. Teachers completed all other assessments. ^d^ Our assessment of maintenance focused on quantitatively and qualitatively assessing teachers’ intentions to continue to use and implement components of the program in the future.

**Table 3 nutrients-17-01922-t003:** *More PEAS Please!* teacher demographics.

Characteristics	Kick-Starter Workshop ^a^		Program Implementers ^b^	
	*n* (%)	M (SD)	*n* (%)	M (SD)
Age	—	43.45 (12.19)	—	43.56 (11.89)
Sex				
Male	0 (0)	–	0 (0)	–
Female	42 (100)	–	24 (100)	–
Race				
Black/African American	30 (81.1)	—	21 (87.5)	—
Hispanic	12 (4.4)	—	0 (0)	—
White	5 (13.5)	—	1 (.04)	—
Asian	1 (2.7)	—	1 (.04)	—
Other	1 (2.7)	—	1 (.04)	—
Ethnicity				
Non-Hispanic/Latino	36 (85.7)	—	23 (95.7)	—
Hispanic/Latino	1 (2.4)	—	0 (0)	—
Prefer not to answer	5 (11.9)	—	1 (4.2)	—
Education ^c^				
Master’s degree or some graduate-level work	4 (9.5)	–	1 (4.3)	–
Bachelor’s degree	13 (31)	–	6 (26.1)	–
Associate’s degree	20 (47.6)	–	12 (52.2)	–
Other	5 (11.9)	–	5 (17.4)	–
Licensure Status				
Licensed teacher (birth– kindergarten or special education)	7 (16.7)	–	3 (12)	–
Other early childhood credential	20 (47.6)	–	14 (56)	–
None or other licensure type	15 (35.7)	–	7 (32)	–
Years Working in Head Start	—	7.38 (6.96)	–	5.86 (4.81)

^a^ Teachers who attended the Kick-Starter Workshop. ^b^ Teachers who attended the Kick-Starter Workshop and continued with program implementation, completing at least one core module by the end of the study. At post-intervention, 19 teachers had completed all intervention components (45.2% retention rate). ^c^ The majority of teachers (*n* = 24 Kick-Starter Workshop; *n* = 17 Program Implementers) reported that their “highest degree completed” was focused on early childhood education.

**Table 4 nutrients-17-01922-t004:** Teacher-reported delivery of *More PEAS Please!* model science learning activities ^a^ (*n* = 24).

	Activity 1	Activity 2	Activity 3	Activity 4	No Activity
Module	*n* (%)	*n* (%)	*n* (%)	*n* (%)	*n* (%)
Support Learning	23 (95.8)	14 (58.3)	15 (62.5)	17 (70.8)	-
Practice Science	20 (83.3)	17 (70.8)	16 (66.6)	17 (70.8)	-
Engage Senses	16 (66.6)	15 (62.5)	13 (54.2)	15 (62.5)	1(0.04)
Apply Science Talk	16 (66.6)	15 (62.5)	18 (75.0)	14 (58.3)	-

^a^ Percentages are based on the sample of program implementers, not the Kick-Start Workshop sample.

## Data Availability

The data presented in this study are available upon request from the corresponding author. Due to privacy concerns, they are not publicly available.

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
