# Peer review of "More PEAS Please!* Process Evaluation of a STEAM Program Designed to Promote Dietary Quality, Science Learning, and Language Skills in Preschool Children"

_nutrients, 2025, doi:10.3390/nu17111922_

Round 1

Reviewer 1 Report

Comments and Suggestions for Authors

This is an innovative intervention for incorporating nutrition into existing curricula for low-income children. The researchers detail the process evaluation for this intervention and provide good detail for items. However, to improve the understanding and readability of the manuscript there is a need for some clarifications and some better visualization of the data through tables and/or figures. 
Introduction – much of the supporting evidence for the need comes out of the research team which is limited by location. Can you cite any other interventions with Head Start or in early childhood education that would also support the rationale? 
Lines123-135: the needs assessment information has been published so suggest removing these details. 
Line 165: researchers indicate that the online learning modules were on-demand. What was the plan for completing these for teachers. Are these all done prior to implementing lessons in the classroom? 
Line193: Were teachers expected to implement all 16 classroom lessons over the HS school year? 
Line 205: Were teachers provided all materials, including foods (target vegetables) for tasting?
Line 211: Assume skin carotenoid score was an outcome variable for the intervention. Provide this context here so that readers understand why this is included.
Line 213: here or in a place appropriate for the manuscript, provide some data on how many teachers participated per center involved. If these LC were center-based, how large were these LC? 
Table 1: The content in this table could be improved with better organization. Perhaps having multiple columns with column headers would improve readability. Also, make sure abbreviations are spelled out.
Figure 3: Is the title of this figure supposed to be “Methods”? The theoretical framework seems like it would fit better in a manuscript related to the curriculum content and outcomes. The researchers don’t seem to use this in this paper for connecting methodology and results (or if they do it is not entirely clear).
Table 2: can the information in the 2nd column be broken into multiple columns to make the information easier to read? There is a column here for maintenance. Was this survey collected within the implementation school year or after this? 
Lines 317-318: can you provide a copy of the interview questions as supplementary materials? 
Lines 357-360: clarification – 8 teachers did not do any module trainings so does that mean that they also did not implement any of the 16 lessons? Was it possible for teachers to implement lessons even if they didn’t do the trainings (If they had access to the materials)?
Lines 398-399: As a reader, I am having some difficulty with timing of assessments. It seems like teachers were asked to complete all modules and then they completed an assessment of this. What was the time frame for teachers to complete these modules? Did they have to have them all completed before they could start to implement lessons? 
Lines 401-405: provide a rationale for recoding from a 5-point likert scale to a scale between 0 to 1. 
General results comment: you present a good deal of data within the text. It seems as though this could be better organized and visualized as a table. This would allow for more clarity on how many people responded to these questions and what their responses were. 
Line 450-454: These appear to be yes/no variables but you are putting them into means/st dev. Can you provide a clearer representation of which learning activities were used by teachers? 
Line 455: You have a small number of participants so use a number instead of a % here.
Line 487: define regularly
Line 515: in this section, can you be more explicit to identify themes that were related to program satisfaction?
Lines 588-593: it seems like the teachers implemented the program over 16 weeks in the spring term. Is this correct? If so, this should be made more clear prior to this point in the manuscript. 
Lines 601-610: Do you describe the methods/scale for contextual factors in the methods? Is this a 5-point likert scale?
Line 629: the researchers bring back the theory and results but these are not explicit in the earlier sections of the manuscript. If there were questions that mapped to the theoretical model, this needs to be more explicit.

Reviewer 2 Report

Comments and Suggestions for Authors

I found this paper to be very interesting, and it is a well-organized study. Nevertheless, it would be advisable to propose a compelling reason for the readers to read this study, clearly articulated within the introductory section. For instance, how does this study differ from previous research and how does it extend that literature?

From an academic perspective, the paper often uses the first person (e.g., I). The third person is to be used by authors in their papers.
